# The Splicing Factor SF2 Is Critical for Hyperproliferation and Survival in a TORC1-Dependent Model of Early Tumorigenesis in *Drosophila*

**DOI:** 10.3390/ijms21124465

**Published:** 2020-06-24

**Authors:** Malgorzata Maria Parniewska, Hugo Stocker

**Affiliations:** Institute of Molecular Systems Biology, ETH Zürich, Otto-Stern-Weg 3, 8093 Zürich, Switzerland; malgorzata.parniewska@ki.se

**Keywords:** TORC1, cancer, tumor suppression, SF2, *Drosophila melanogaster*, Pten

## Abstract

The Target of Rapamycin complex 1 (TORC1) is an evolutionarily conserved kinase complex coordinating cellular growth with nutritional conditions and growth factor signaling, and its activity is elevated in many cancer types. The use of TORC1 inhibitors as anticancer drugs is, however, limited by unwanted side-effects and development of resistance. We therefore attempted to identify limiting modulators or downstream effectors of TORC1 that could serve as therapeutic targets. *Drosophila* epithelial tissues that lack the tumor suppressor Pten hyperproliferate upon nutrient restriction in a TORC1-dependent manner. We probed candidates of the TORC1 signaling network for factors limiting the overgrowth of *Pten* mutant tissues. The serine/arginine-rich splicing factor 2 (SF2) was identified as the most limiting factor: *SF2* knockdown drives *Pten* mutant cells into apoptosis, while not affecting control tissue. SF2 acts downstream of or in parallel to TORC1 but is not required for the activation of the TORC1 target S6K. Transcriptomics analysis revealed transcripts with alternatively used exons regulated by SF2 in the tumor context, including *p53*. SF2 may therefore represent a highly specific therapeutic target for tumors with hyperactive TORC1 signaling.

## 1. Introduction

Target of Rapamycin complex 1 (TORC1) integrates inputs such as growth factor signaling and nutritional status with cellular growth. It functions as a downstream effector of the key oncogenic pathways PI3K/AKT and Ras/MAPK. Accordingly, TORC1 is hyperactivated in many cancer types [1,2,3]. The regulation of growth by TORC1 is mainly attributed to the control of protein synthesis, but TORC1’s growth-promoting functions extend well beyond protein synthesis and include lipid and nucleotide synthesis, glutaminolysis, aerobic glycolysis or splicing [1,4].

Attenuation of TORC1 activity as a means to combat cancer has been widely explored by the use of TORC1 inhibitors such as rapamycin and rapamycin-based analogs (rapalogs) [5]. However, the efficacy of such treatments has been limited by the complexity of the TORC1 signaling network. For example, a negative feedback loop from the TORC1 downstream target S6K attenuates PI3K signaling upon TORC1 activation [6,7]. Thus, targeting modulators or effectors of TORC1 may be a more promising approach than targeting TORC1 itself. In recent years, high-throughput technologies enabled a significant expansion of the TORC1 signaling network by generating vast datasets of novel TORC1 regulators and targets [8,9,10,11]. These components could serve as new targets, but many of them still require an in vivo validation, particularly in the context of tumorigenesis. 

We have established a model for early tumorigenesis in *Drosophila*, to validate putative modulators and effectors of TORC1 activity in vivo. Early stages of tumorigenesis are mimicked in larval imaginal discs, single-cell-layered epithelia that will give rise to adult structures such as legs, wings and eyes. Pten (phosphatase and tensin homolog deleted on chromosome 10), which functions by antagonizing PI3K, is one of the most commonly mutated tumor suppressors in human cancers [12] and is well-conserved in *Drosophila* [13,14,15]. Clones of *Pten* mutant cells generated in eye imaginal discs are enlarged and hyperproliferate under conditions of nutrient restriction (NR) [16]. This behavior is consistent with mammalian models where tumors carrying PI3K-activating mutations have been shown to be resistant to conditions of NR [17]. As the hyperproliferative behavior of *Pten* mutant tissues is strictly dependent on TORC1 activity [16], we exploited the *Pten* phenotype to test candidates from the TORC1 signaling network by RNAi for their requirement for the overgrowth. We show that the SR-rich Splicing factor 2 (SF2), previously identified as a phosphorylation target of TORC1 [11], is critical for the overgrowth of *Pten*-deficient tissues. 

## 2. Results

### 2.1. Screen for TORC1 Signaling Components Limiting the Overgrowth of Pten-Deficient Epithelia under NR 

*Pten*-deficient tissues in *Drosophila* massively hyperproliferate upon NR in a TORC1-dependent manner (henceforward called *Pten* overgrowth) [16]. To establish a system that does not depend on homozygous *Pten* mutant cells generated by mitotic recombination, we tested whether eye-specific knockdown of *Pten* results in a similar overgrowth phenotype upon NR. We used *eyFlp* to tissue-specifically excise the FRT cassette of an *Act>CD2>Gal4* transgene, resulting in Gal4 expression in all cells of the eye imaginal disc. Driving *UAS-Pten-RNAi* during eye development resulted in a size increase of 31% upon normal feeding conditions. This size increase was enhanced to 49% upon NR (compared to control eyes on normal conditions, Figure 1a). Thus, instead of the normal reduction of tissue size upon NR, *Pten* knockdown eyes overgrow upon NR (with a ratio of 1.14, i.e., 14% overgrowth). Our previous work indicated that the *Pten* overgrowth fully depends on TORC1 activity [16]. We therefore reasoned that this system could be used to screen for components of the TORC1 signaling network that are limiting for the *Pten* overgrowth upon NR.

We performed RNAi-mediated double knockdowns of *Pten* and candidate genes, in a Gal4/UAS-dependent manner, specifically in the eye imaginal discs. Larvae were subjected to normal and NR conditions during development, and the size of the adult eye was used as readout (Figure 1b). Candidate genes were chosen from four published genome- and proteome-wide screens that led to the identification of novel regulators and effectors of TORC1 activity in *Drosophila* S2 cells and in mammalian cells. The identified components regulate TORC1 as assayed by the phosphorylation of S6 (direct target of S6K) [8] or are affected by TORC1 inhibition at the level of transcription [9] or phosphorylation [10,11]. Most of these candidates have not been functionally tested for a putative role in tumorigenesis so far. We applied stringent selection criteria to narrow down the set of candidates (Appendix A), resulting in 294 candidates, of which 256 have homologs in *Drosophila* (Appendix A). We selected 296 RNAi lines from the Vienna Drosophila Resource Center (VDRC) collection targeting these candidate genes.

Twenty of the 296 knockdowns resulted in suppression of the *Pten* overgrowth (Figure 1c, Appendix A). The eye sizes were normalized to the *Pten* knockdown under normal conditions, and the suppressive effects were ranked according to the ratio of eye size under NR to the eye size under normal conditions (eye 10/eye 100) (Figure 1d,e). The suppressors were also tested in a wild-type background to exclude additive effects (Appendix A). We were especially interested in suppressors of the *Pten* overgrowth upon NR with little effect on the *Pten* phenotype upon normal conditions and no effect in a wild-type background. Only two RNAi lines fulfilled all criteria and strongly suppressed the *Pten* overgrowth, both targeting the *serine-arginine rich splicing factor 2 (SF2)*. Whereas they resulted in the strongest suppression of the *Pten* overgrowth, they did not affect eye size in a control background (Figure 1f). SF2 is the *Drosophila* homolog of SRSF1 (also known as alternative splicing factor 1 (ASF1) or pre-mRNA-splicing factor SF2), which was identified as a putative phosphorylation target of TORC1 [11].

Three lines of evidence support the specificity of the suppressive effect. First, the two RNAi lines against *SF2* fully rescued a previously reported *SF2* overexpression phenotype in eye tissues (Appendix A). Second, the knockdown resulted in decreased SF2 protein levels as assessed by an SF2-specific antibody staining (Appendix A). Third, independent RNAi lines targeting different sequences of *SF2* also suppressed the *Pten* overgrowth (Appendix A).

SF2 belongs to the conserved family of SR-rich proteins involved in RNA splicing. The human SR protein family consists of 12 members, eight of which are conserved in *Drosophila* (SF2, SC35, Srp54, 9G8, Rbp1, B52, Rsf1 and Rbp1-like). To test whether the suppression of *Pten* overgrowth is specific to SF2, all SR protein family members were co-knocked down with *Pten*. With the exception of RNAi targeting *B52* and *SRp54*, which resulted in strong malformations of the eye and pupal lethality, respectively, knocking down other SR protein genes did not alter the *Pten* overgrowth (Appendix A). Therefore, SF2 plays a critical and specific role in the overgrowth of *Pten* mutant tissues upon NR.

### 2.2. The Splicing Factor SF2 Is Required for Maintaining Pten Mutant Cells within the Epithelium

The assessment of the candidate gene knockdowns was performed in the adult eye. To study the phenotype during the proliferative phase, eye imaginal discs of *Pten* and *SF2* co-knockdown under normal and NR conditions were dissected and stained for the apoptotic marker cleaved Dcp-1. Whereas *Pten*-deficient eye imaginal discs increased in size throughout development under NR, discs deficient for both *Pten* and *SF2* were smaller already at an early stage and throughout the proliferative stage (Figure 2). The size decrease is likely a result of increased apoptosis, as apoptotic cells accumulated throughout the growth period. The co-knockdown on normal feeding conditions also resulted in an increase of apoptotic cells, but the size of the discs remained unaffected.

To mimic the clonal nature of cancer, we analyzed the requirement of SF2 in a clonal context. We used the twin MARCM (Molecular Analysis with a Repressible Cell Marker) technique, which allows for the generation of cells homozygous mutant for *Pten* and expressing any gene of interest under Gal4/UAS-control within an otherwise heterozygous tissue. The method enables tracking of clones of *Pten* homozygous mutant cells and of the corresponding sister clones (twinspots) of cells carrying two *Pten* wild-type copies, based on combined positive and negative labeling with genetically encoded markers (see Material and Methods).

Clones of *Pten* mutant cells (*Pten* clones) hyperproliferate upon NR, overtaking most of the eye imaginal disc and strongly reducing the twinspots ([16]; Figure 3a, upper panel). Knocking down *SF2* within the *Pten* clones upon NR completely abrogated the overgrowth and resulted in massive apoptosis of the clones (Figure 3a, lower panel). The dying clones were no longer maintained within the epithelial layer but were extruded through the basal side of the disc (“basal” in Figure 3a). Whereas the knockdown of *SF2* did not visibly affect the growth of *Pten* mutant cells under normal conditions in the (non-clonal) screening system, it caused the elimination of the *Pten* clones (although with less apoptotic signals) under normal feeding conditions (Figure 3b, lower panel). Clonal knockdown of *SF2* in a wild-type background caused single apoptotic cells but did not visibly impact the growth of the clones (Figure 3c). These observations were also reproduced in wing imaginal discs, excluding tissue-specific effects (Appendix A). Thus, SF2 function is required to maintain *Pten* clones in a growing epithelium.

Since mammalian SF2 is a proto-oncogene capable of transforming rodent fibroblasts upon overexpression [18], we tested whether *SF2* overexpression causes overgrowth in *Drosophila* imaginal tissues. Rather than inducing overgrowth, both ubiquitous and clonal overexpression of *SF2* triggered apoptosis of the tissue (Appendix A). The observed apoptosis is likely a consequence of toxic SF2 levels achieved by Gal4/UAS-mediated overexpression.

### 2.3. SF2 Acts Downstream of or in Parallel to TORC1 and Is Required for Survival of Tsc1 Mutant Clones as Well

SR-rich splicing factors possess an RRM (RNA recognition motif) at the N-terminus and an SR-repeat-rich domain at the C-terminus. Yu et al. [11] identified two phosphorylation sites located at the C-terminal, SR-repeat-rich end of SRSF1, whose phosphorylation was decreased by TORC1 inhibition. However, these sites are not conserved in *Drosophila*. It is therefore unclear whether SF2 is also regulated in response to TORC1 activity in the fly. SF2 protein levels remained unchanged in *Pten* mutant clones, both under normal and NR conditions (Appendix A). To assess whether SF2 is required for phenotypic effects caused by TORC1, we performed a genetic epistasis analysis with Rheb, the small GTPase directly involved in TORC1 activation. The overgrowth and malformations of the adult eye caused by *GMR-Gal4*-driven overexpression of *Rheb* in the differentiating cells of the eye imaginal discs were partially suppressed by the knockdown of *SF2*, suggesting that SF2 acts downstream of or in parallel to TORC1 (Appendix A). 

Overexpression and knockdown of *SRSF1* have been shown to increase and decrease, respectively, the phosphorylation of the TORC1 targets S6K and 4E-BP1 in various immortal and primary cell lines [19]. We therefore tested whether SF2 impacts the high S6K activity in imaginal disc cells with elevated TORC1 activity (achieved by the loss of a negative regulator of TORC1, Tsc1). Clones of *Tsc1* mutant cells displayed high levels of S6 phosphorylation as detected by a phospho-S6 antibody, indicative of increased TORC1 and S6K activities [20] (Figure 4a,c). Knockdown of *SF2* did not affect the high phospho-S6 signal in *Tsc1* mutant cells. Upon NR, the *SF2* knockdown triggered apoptosis in *Tsc1* mutant cells (Figure 4b,d). Thus, SF2 is required for the survival of both *Pten* and *Tsc1* mutant cells upon NR. It acts downstream of or in parallel to TORC1 but is not required for S6K activity.

### 2.4. Splicing Targets of SF2 in Pten-Deficient Tissues under NR

SR proteins regulate both constitutive and alternative splicing by recognizing distinct splice sites on pre-mRNAs [21]. Since *SF2* knockdown specifically suppresses the massive hyperproliferation of *Pten*-deficient tissue upon NR, we set out to identify the splicing targets of SF2 in this tumorigenic context by RNA-seq. 

Eye imaginal discs dissected from mid-L3 larvae with *Pten* and *SF2* co-knockdown under normal and NR feeding conditions were subjected to RNA-seq. Eye imaginal discs with *SF2* knockdown alone, under normal conditions, were also included in the analysis to identify SF2 targets in control tissue. The small size of control discs upon NR precluded their inclusion in the RNA-seq analysis. Both *Pten* and *SF2* mRNA levels were significantly decreased upon the respective RNAi (Appendix A). No change in *SF2* mRNA abundance was observed in *Pten*-deficient tissues upon NR, consistent with the protein expression (Appendix A).

We first analyzed the RNA-seq data for changes in transcript abundance. Differential expression analyses using edgeR (Appendix A) revealed 533 differentially expressed genes (DEGs) upon *SF2* knockdown in *Pten*-deficient discs from NR conditions (Figure 5a), 393 DEGs in *Pten*-deficient discs from normal conditions and 22 DEGs in the *SF2* knockdown alone (Appendix A) (with a log_2_FC 0.5 and FDR <0.01). Although 362 DEGs were unique to the co-knockdown upon NR (Figure 5b), changes shared with other conditions may also become limiting specifically under conditions of both *Pten* deficiency and NR.

GO terms enrichment analysis of DEGs changing with *SF2* knockdown in combination with *Pten*-deficiency and NR (all DEGs, FDR <0.01) revealed a strong enrichment of genes belonging to the GO categories “glutathione metabolic process” and “oxidation-reduction process” (Figure 5c). These include detoxifying enzymes, glutathione S transferases (GSTs) together with glutamate-cysteine ligase (*Gclc*), a rate-limiting enzyme in glutathione (GSH) synthesis and cytochrome P450 genes (*Cyp* genes) (Figure 5d,e), the majority of which are upregulated in the co-knockdown upon NR. GST enzymes have been shown to interact with JNK to inhibit apoptosis in stress-inducing conditions [22]. However, we found a slight upregulation of the JNK downstream components *Mmp1*, *puc* (FDR = 0.04) and *kay* (Figure 5f). Possibly the upregulation of detoxifying enzymes reflects a failed attempt to rescue the tissue from undergoing apoptosis caused by the knockdown of *SF2*. It is therefore conceivable that many of the observed changes in transcript abundances are indirect consequences of the loss of *SF2*.

Since the oncogenic properties of SRSF1 have been linked to its splicing activity, we used DEXSeq, a software to test for differential exon usage from RNA-seq data, to identify SF2 target exons in *Pten*-deficient tissues upon NR [23] (Appendix A). At a FDR of 0.1, 903 significantly changing exons were detected in the *Pten* and *SF2* co-knockdown upon NR (compared to *Pten* knockdown alone) (Figure 6a), 223 exons in the *Pten* and *SF2* co-knockdown under normal conditions, and 919 exons in the *SF2* single knockdown (compared to wild-type control) (Appendix A). There was only a small overlap between DEGs and genes with changing exons detected by DEXSeq (Appendix A), suggesting that most of the genes with exon level changes are specifically regulated by SF2’s splicing activity without an impact on overall mRNA abundance, consistent with the notion that many of the DEGs are secondary effects of the *SF2* knockdown.

We focused our analysis on exons where the *SF2* knockdown resulted in an at least two-fold change in expression compared to control conditions. We detected 86 exons (Figure 6a) corresponding to 63 genes changing in the *Pten* and *SF2* co-knockdown upon NR, 28 exons corresponding to 21 genes changing in the *Pten* and *SF2* co-knockdown under normal conditions and 35 exons corresponding to 29 genes in the *SF2* single knockdown (Appendix A). Twenty of the 86 most changing exons in the *Pten SF2* co-knockdown upon NR were shared with the *Pten SF2* co-knockdown under normal conditions and/or *SF2* knockdown alone (Figure 6b). These exons are likely general SF2 targets. Although not specific to the *Pten SF2* co-knockdown upon NR conditions, these differential exon usage events may still be critical for the behavior of *Pten*-deficient tissues upon NR.

We analyzed which types of splicing events were affected by *SF2* knockdown. Interestingly, the *SF2* knockdown impacted the exon usage of its own mRNA (Figure 6c). A similar autoregulatory function has also been observed for SRSF1 in Hela cells [24]. The *SF2* mRNA preferentially expressed upon *SF2* knockdown carries an extended 3’ UTR (*SF2-RB* variant). A transcript with longer 3′ UTR is also more abundant after *SF2* knockdown in the case of *vielfaltig* (also known as *zelda*), which encodes a zinc finger transcription factor. These changes are consistent with published data showing the involvement of SF2 in 3′ end processing of target transcripts, in addition to regulating exon inclusion and repression events [21]. *AKAP200* mRNA displayed cassette exon regulation with a >2-fold upregulated inclusion of counting bin E008 upon *SF2* knockdown. This sequence has been reported to be present in the longer variant of the gene [25]. SF2 also regulated the use of mutually exclusive exons, as detected by a switch in isoform abundance of *Traf4*, the *TNF-receptor-associated factor 4*, from *Traf4-RE* being more abundant in *Pten*-deficient control tissues upon NR to *Traf4-RA* upon *SF2* knockdown. Finally, we also detected differential regulation of transcription start sites by SF2. These include a change in abundance of a *CD98hc* (encoding the amino acid transporter SLC32A) transcript variant with an extended 5′ UTR and *p53*, where the transcript variant *p53E* starting at E005 is upregulated upon *SF2* knockdown (with an FDR of 0.2).

## 3. Discussion

The TORC1 signaling network has been substantially expanded by recent research, providing numerous potential targets for therapeutic intervention in cancers with elevated TORC1 activity. However, in vivo assessments of the requirement of the new candidates for tumor development are often lacking. We have exploited the genetically amenable model organism *Drosophila* to identify TORC1 signaling network members that are limiting in a model of early tumorigenesis. Epithelial tissues deficient for the tumor suppressor Pten hyperproliferate upon NR in a strictly TORC1-dependent manner [16]. We show here that the *Pten* mutant cells critically depend on the splicing factor SF2: *Pten* mutant cells with a knockdown of *SF2* trigger apoptosis and are extruded from the epithelium (Appendix A). 

SF2 is the *Drosophila* homolog of the proto-oncogene SRSF1. Silencing of *SF2* has been shown to increase the number of G2/M cells and the length of G2/M phase in cultured *Drosophila* cell lines [26] and in eye and wing imaginal discs [27]. It is therefore likely that the apoptosis of *Pten* mutant cells with an *SF2* knockdown is linked to a cell-cycle arrest at G2/M.

*SRSF1* mRNA levels are increased in many cancers [28,29,30,31]. We did not observe any change in *SF2* abundance upon loss of Pten, neither at the transcript nor at the protein level. It is possibly the activity of SF2 that responds to the Pten status (and therefore TORC1 activity). SF2 was chosen as a candidate based on an analysis of the TORC1-dependent phosphoproteome, where SRSF1 phosphorylation was detected at two sites in its RS domain [11]. SRSF1 is not phosphorylated directly by TORC1 but by SRPK2, which is directly phosphorylated by S6K upon TORC1 activation [4]. *Drosophila* SF2 lacks eight SR repeats in its RS domain, and the two TORC1-dependent phosphosites are not conserved. It remains to be determined whether SF2 is phosphorylated at other sites in response to TORC1 activity. We have tested all SRPK homologs in our assay, and RNAi lines corresponding to SRPK and SRPK79D indeed did suppress the *Pten* overgrowth (the eye 10/eye 100 ratios were 0.99 for SRPK and 1.01 for SRPK79D, respectively). Therefore, although the exact mode of SF2 regulation in *Drosophila* remains to be determined, its activity is likely to be controlled via the TORC1-SRPK axis. 

Changes in transcript levels and splicing patterns upon knocking down *SF2* in *Pten*-deficient tissues under normal and NR conditions were analyzed by RNA-seq. We detected altered abundances of genes involved in glutathione metabolism including *Gclc*, the rate-limiting enzyme of glutathione biosynthesis, and GSTs, which catalyze the conjugation of reduced GSH to its substrates. GSH has been shown to serve a protective role against apoptosis, by modulating JNK activity via GSTs in response to oxidative stress [22]. Indeed, the downstream targets of JNK *kay*, *puc* and *mmp1* were upregulated selectively in the *SF2* knockdown in *Pten*-deficient tissues upon NR, suggesting induction of oxidative stress. However, NADPH oxidase (Nox), a marker of oxidative stress, is strongly downregulated under these conditions. The upregulation of GSTs was accompanied by the upregulation of Cytochrome P450s. GSTs and Cyp 450s have been shown to be co-upregulated in response to xenobiotic treatment in *Drosophila* [32]. We speculate that other metabolic changes, rather than oxidative stress, induce detoxification mechanisms in an attempt to cope with the massive cell death resulting from *SF2* knockdown in *Pten*-deficient tissues upon NR.

The tumorigenic potential of mammalian SRSF1 is attributed mainly to its splicing activity. Using the DEXSeq software, we detected a relatively low number (86 exons) of exons strongly induced or repressed (|log_2_FC| 1) upon knocking down *SF2* in *Pten*-deficient tissues under NR. These transcripts show little overlap with DEGs and therefore are regulated by SF2 exclusively via splicing and 5′ or 3′ UTR selection, as reported previously [21]. SRSF1 targets are context-dependent with different substrates identified across breast, lung and colon cancer [29,30]. Accordingly, we did not find much overlap with targets identified in previous studies, further demonstrating that SF2 displays selectivity toward defined mRNAs under different conditions. The splicing changes occurring in our model point to specific isoforms that are presumably required to cope with the loss of Pten upon NR. Some of these exon usage changes have been described in various growth-regulating contexts previously. The *zelda-RD* transcript encodes a shorter isoform lacking three of the four zinc fingers. This short isoform blocks the activation of transcription by the full-length protein in the early embryo in a dominant-negative manner [33]. The *zelda-RD* transcript is more abundant in *Pten* deficient tissues upon *SF2* knockdown. The *Drosophila* genome encodes four p53 isoforms (A, B, D and E) with different apoptotic potentials. Whereas p53A is the primary mediator of the apoptotic response to DNA damage, p53E has been shown to possess a protective function against genotoxic stress ([34]. Increased levels of *p53E* in *Pten*- and *SF2*-deficient tissues upon NR suggest a context-dependent, dominant negative function that, likely together with other SF2 targets, might mediate the suppression of *Pten* overgrowth upon NR. Based on the different biological functions of the SF2 targets, we hypothesize that SF2 does not control a sole biological process, allowing *Pten*-deficient tissues to hyperproliferate upon NR. Instead, SF2 controls various transcripts giving rise to proteins involved in diverse processes, which, most likely in combination, are necessary to govern the overgrowth phenotype.

Therapeutic approaches for targeting splicing in cancer include the development of splice variant-specific siRNAs, splice switching antisense oligonucleotides (SSOs) that bind to specific sites on the pre-mRNA and block the recruitment of splicing factors, thereby preventing exon exclusion or inclusion events that result in disease-specific transcripts [35], and small molecule compounds that inhibit either components of the spliceosome or splicing factor kinases [36,37]. Several small molecules inhibiting splicing are currently under clinical trials, but some cause toxicity resulting from widespread splicing alterations that can lead to the accumulation of unspliced pre-mRNAs in the nucleus and cell cycle arrest [38]. In this light, the SF2-regulated alternatively spliced transcripts that we have identified are promising candidates for specific transcript-based therapy.

We have identified SF2, the *Drosophila* homolog of human *SRSF1*, as an Achilles heel of pre-tumorous cells devoid of the tumor suppressors *Pten or Tsc1*, which rely on TORC1 activity to grow and proliferate. *Pten* and *Tsc1* mutant cells are addicted to the function of SF2; silencing *SF2* drives them into apoptosis. As control cells are hardly affected by *SF2* silencing, SRSF1 represents a promising target for cancer therapy.

## 4. Materials and Methods 

### 4.1. Fly Media and Maintenance

*Drosophila melanogaster* were reared on standard medium containing 75 g sugar, 55 g cornmeal, 10 g wheat flour, 8 g agar, 100 g fresh yeast and 15 mL *v/v* nipagin in 1 L water. Nutrient restriction (NR) food was generated by decreasing the yeast content to 10 g/L, without altering the amount of the other ingredients. Crosses and egg layings (EL) were performed at 25 °C and 70% relative humidity.

### 4.2. Preparation of Crosses for Screening

Approximately 50 virgins of the *y w eyFlp Act>CD2>Gal4; UAS-Pten-RNAi* tester line and 15 males of the *UAS*-candidate-RNAi lines were used for each cross. Crosses were set up in standard culture vials, with normal food, and the flies were allowed to mate for two days. The crosses were then transferred to egg-laying cages, with apple agar plates and a drop of yeast paste, for another two days. The final egg laying was done for 24 h. The embryos were then collected, cleaned with water by filtration and spread to a minimum of 3 vials of 100 g/L food and 4 vials of 10 g/L food. A predefined pile of embryos corresponding to 100 individuals was spread to each vial. Larvae were allowed to develop at 25 °C and 70% relative humidity and screened 14 days after egg spreading.

### 4.3. Eye Size Measurement and Quantification

Twelve females per genotype and feeding condition were collected and frozen. Eye sizes were quantified by using an in-house software Flyeyeball, which measures the area of a region of interest based on color thresholding. Scoring of the phenotypes was based on the size difference between eyes under the two different conditions expressed as the ratio of eye size upon NR to the eye size under normal conditions (eye 10/eye 100). A threshold of 1 (identical eye size under both feeding conditions) was applied as a cutoff value for defining the suppressors.

### 4.4. Mutants, Transgenes and Crosses

Homozygous *Pten^117^* clones were generated by using *y w hsFlp*; *FRT40 ubi-GFP tub-Gal80*; *UAS-myr-RFP tub-Gal4* (gift from Elisabeth Fischer) in combination with the *FRT40 Pten^117^* allele (or *FRT40iso*) recombined with *UAS*-control-RNAi (VDRC line v47096, control) or *UAS-SF2-RNAi* (VDRC line v27775) transgenes. Homozygous *Tsc1* mutant clones were generated by using *y w hsFlp*;; *FRT82 ubi-GFP* in combination with the *FRT82 Tsc1^Q87X^* allele combined with the aforementioned *UAS*-control-RNAi or *UAS-SF2-RNAi* transgenes on the second chromosome. Clones were generated by heat shocking larvae for 15 min, at 36 h after egg laying (AEL). Discs were dissected 72 h after heat shock for larvae developing on normal food, and 120 h after heat shock for larvae developing on NR food. Clones bearing control knockdown (*UAS*-control-RNAi), *SF2* knockdown (*UAS-SF2-RNAi*), control overexpression (*UAS-lacZ*) and *SF2* overexpression (*UAS-SF2*) were generated by using *y w hsFlp*;; *Act>CD2>Gal4 UAS-GFP*/*TM6B* and *y w hsFlp*; *UAS-Pten-RNAi*; *Act>CD2>Gal4 UAS-GFP*/*TM6B*. Clones were generated by heat shocking larvae for 12 min at 36 h AEL. Discs were dissected 72 h after heat shock for larvae developing on normal food and 120 h after heat shock for larvae developing on NR food. Knockdown of *Pten* was achieved based on the VDRC line v101475, using the *y w eyFlp Act>CD2>Gal4*; *UAS-Pten-RNAi* tester line. Candidate genes of interest were knocked down using lines from the BDSC, VDRC and NIG-Fly stock centers listed in Appendix A. Moreover, *y w; GMR-Gal4*/*CyO y+*; *GMR-Rheb*/*TM6B* was used to test for suppression of the eye phenotype caused by activated TORC1. *GMR-Gal4 UAS-dASF* (gift from Francois Juge) was used as the basis for *UAS-SF2*.

### 4.5. Immunohistochemistry and Image Acquisition

Imaginal discs were dissected in ice-cold 1xPBS, fixed in 4% PFA for 30 min, at RT, and permeabilized in 0.3% PBT (PBS with Triton-X100). Nuclei were stained with DAPI (1:2000). Apoptosis was visualized, using anti-cleaved-Dcp1 antibody (Cell Signaling) in 2% NDS (1:100). The anti-SF2 antibody was a gift from Francois Juge and used at 1:1000 in 2% NDS. Primary antibodies were detected by using goat anti-rabbit Alexa633 fluorophores. A Leica SPE confocal laser scanning microscope was used for image acquisition. Adult fly eyes were imaged, using a KEYENCE VHX100 digital microscope.

To check the stage at which SF2 is required for the *Pten* overgrowth, discs from normal conditions were dissected at 96 h (mid-L3 larvae) and 120 h (late-L3 larvae) AEL. Since the larvae developing on NR conditions are approximately 2 days delayed compared to their normally fed siblings, these larvae were dissected at 120, 144, 156 and 168 h AEL.

### 4.6. Statistical Analysis

Student’s *t*-test (two-tailed) was used to test for significance. Twelve individuals were measured for each genotype, unless otherwise indicated in the graphs. Significance is indicated by * (*p* < 0.05), ** (*p* < 0.01), *** (*p* < 0.001) and **** (*p* < 0.0001). Error bars represent standard deviation.

### 4.7. RNA Isolation, Library Preparation and Sequencing

Eye imaginal discs were dissected in ice-cold PBS. As larval development is delayed and the time window of pupariation initiation massively extended upon NR, the time points for dissection had to be carefully evaluated. The position of the morphogenetic furrow served as criterion to select larvae at the same developmental stage. Discs from larvae reared on normal food were dissected 96 h AEL (mid-L3), whereas discs from larvae on NR were dissected 156 h AEL. Twenty discs were used for all samples, except for the co-knockdown of *Pten* and *SF2* upon NR (PTEN RNAi SF2 RNAi 10), where 60 discs were used. Due to the small disc size at the chosen time points, not enough material was obtained for the *SF2* and control knockdowns in a wild-type background upon NR. Samples were prepared in triplicates (resulting in a total of 18 samples) and immediately frozen at −80 °C, until RNA extraction. Total RNA was extracted by using QIAGEN RNeasy Plus Micro kit according to the manufacturer’s protocol. Poly(A) enrichment using SmartSeq2 [39] of mRNAs, library preparation and sequencing were performed at the Functional Genomics Center, University of Zürich, Switzerland. Libraries were sequenced on the Illumina HiSeq 2000 platform, generating 150 bp paired-end reads.

### 4.8. Read Mapping, Differential Gene Expression and Alternative Splicing Analysis

Read quality was assessed by using FastQC. Reads were then mapped to the reference genome BDGP6 (R6.89), using Tophat2. We used edgeR [40] to test for DEGs upon *SF2* knockdown. DEXSeq, a software to test for differential exon usage from RNA-seq data [23], was used to detect genes with splicing changes upon *SF2* knockdown. PCA analysis of all 18 replicates revealed a separate cluster for three replicates of different samples (Appendix A). We therefore decided to exclude these replicates from the edgeR and DEXSeq analyses. Data have been deposited in the European Nucleotide Archive (ENA, accession PRJEB39006).

### 4.9. GO Term Enrichment Analysis

DAVID was used for GO term enrichment, term Biological Process. DEGs and AS genes were used as foreground, and all FlyBase annotated genes (FB2018_05) as background.

## Figures and Tables

**Figure 1 ijms-21-04465-f001:**
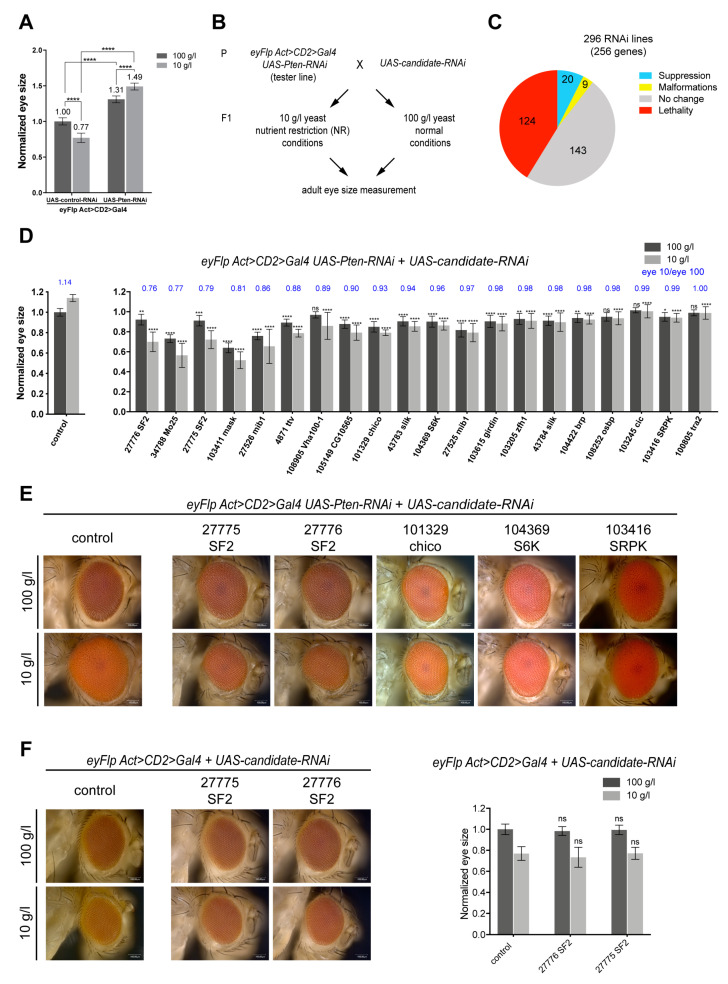
Screen for TORC1 signaling components limiting for the *Pten* overgrowth phenotype. (**A**) Eye-specific *Pten* knockdown enlarges eye size more strongly under NR, compared to normal conditions (*Pten* overgrowth). (**B**) Experimental design of the candidate screen. (**C**) Results from all 296 RNAi lines tested. (**D**) Normalized eye size after candidate knockdown together with corresponding eye 10/eye 100 ratios (blue). The *eyFlp Act>CD2>Gal4; UAS-Pten-RNAi* tester line crossed to a control RNAi line (targeting *CG1315*) yields a ratio of 1.14, reflecting the overgrowth upon NR. The smaller the eye 10/eye 100 value, the better the suppression. An eye 10/eye 100 value of 1.00 was chosen as threshold for suppressors. Since candidates were tested in several batches, the control represented on the left is a chosen representative. The eye size of each candidate was normalized to the control on normal conditions from the respective batch. Student’s *t*-test was used to compare eye sizes after candidate knockdown to eye size of control on corresponding food condition with: * *p* < 0.05, ** *p* < 0.01, *** *p* < 0.001, **** *p* < 0.0001, ns: not significant. Error bars represent standard deviation. (**E**) Eyes of selected suppressors. (**F**) Eyes of flies with *SF2* knockdown in control tissue with quantifications.

**Figure 2 ijms-21-04465-f002:**
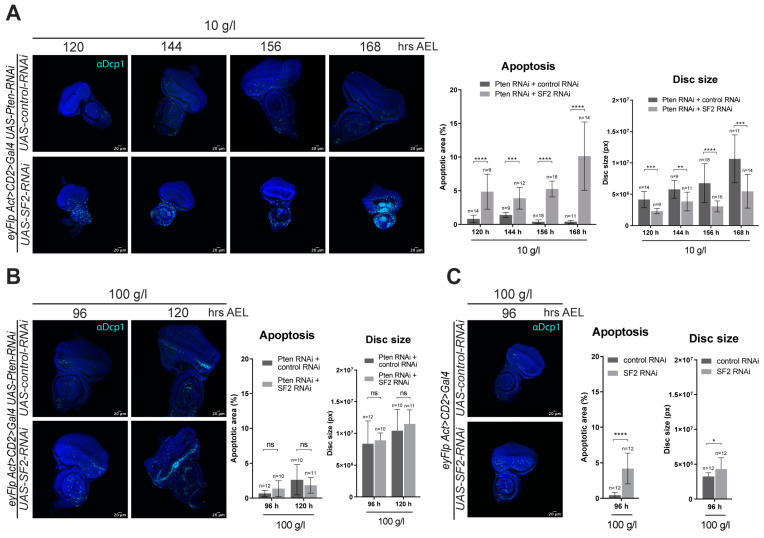
*SF2* knockdown suppresses the growth of *Pten*-deficient eye imaginal discs. The knockdowns were achieved by using the *eyFlp Act>CD2>Gal4; UAS-Pten-RNAi* tester line. (**A**) Eye discs upon NR were dissected at 120, 144, 156 and 160 h after egg laying (AEL). The growth of *Pten*-deficient discs is blocked by *SF2* silencing. Discs with *SF2* knockdown become increasingly apoptotic, as visualized by cleaved-Dcp1 staining. (**B**) Discs from normal conditions were dissected 96 and 120 h AEL. *SF2* knockdown does not affect the growth of *Pten*-deficient discs under normal conditions. (**C**) *SF2* knockdown also causes some apoptosis in control discs, without affecting growth. * *p* < 0.05, ** *p* < 0.01, *** *p* < 0.001, **** *p* < 0.0001, ns: not significant. Scale bars represent 20 μm.

**Figure 3 ijms-21-04465-f003:**
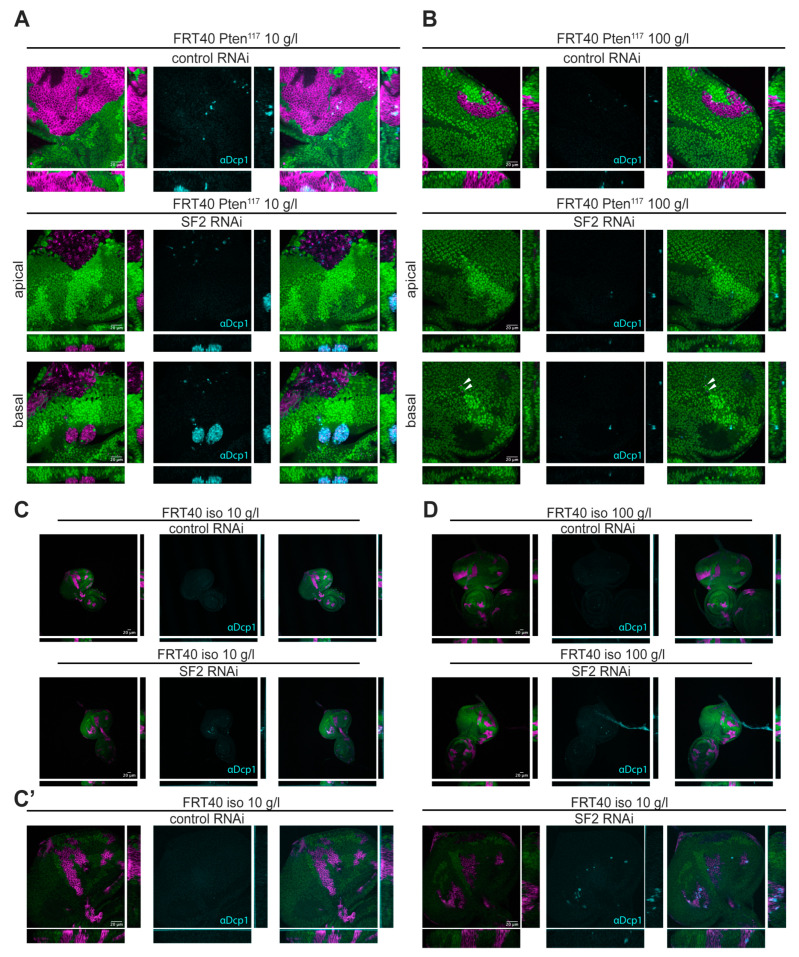
Massive apoptosis in *Pten* mutant cells with *SF2* knockdown (**A**) *SF2* knockdown in *Pten^117^* mutant cells (generated by the twin MARCM technique) results in massive apoptosis of the *Pten* mutant tissue as visualized by cleaved-Dcp1 staining upon NR (lower panel). The affected cells are located anterior to the morphogenetic furrow, induce apoptosis and are extruded from the epithelial layer. (**B**) *SF2* knockdown eliminates *Pten* clones also under normal conditions. (**C**,**C′**) Control (*iso)* clones are only mildly affected by *SF2* knockdown, showing single apoptotic cells under NR. (**D**) Control *(iso)* clones are also mildly affected by *SF2* knockdown under normal conditions. In all panels, clones are marked positively by RFP (magenta) and negatively by the absence of GFP (green). xz- and yz-cross-sections are shown below and on the right, respectively. Scale bars represent 20 μm.

**Figure 4 ijms-21-04465-f004:**
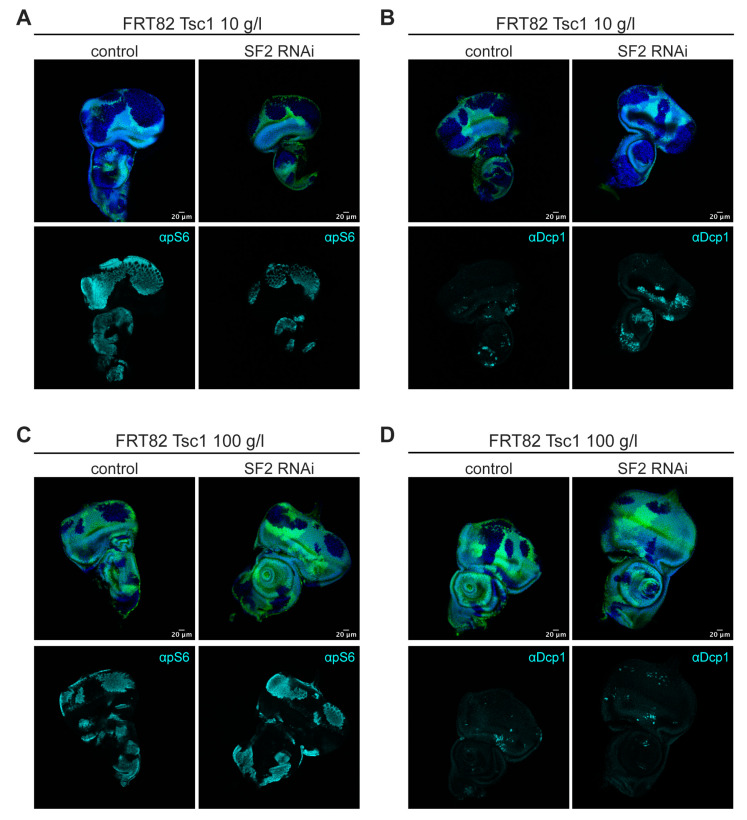
*SF2* knockdown drives *Tsc1* mutant cells into apoptosis but does not affect S6K activity. *SF2* knockdown in *Tsc1* mutant MARCM clones does not affect pS6 levels under normal (**C**) and NR conditions (**A**). *SF2* knockdown in *Tsc1* mutant MARCM clones results in apoptosis specifically upon NR (**B**), but not under normal conditions (**D**). Scale bars represent 20 μm.

**Figure 5 ijms-21-04465-f005:**
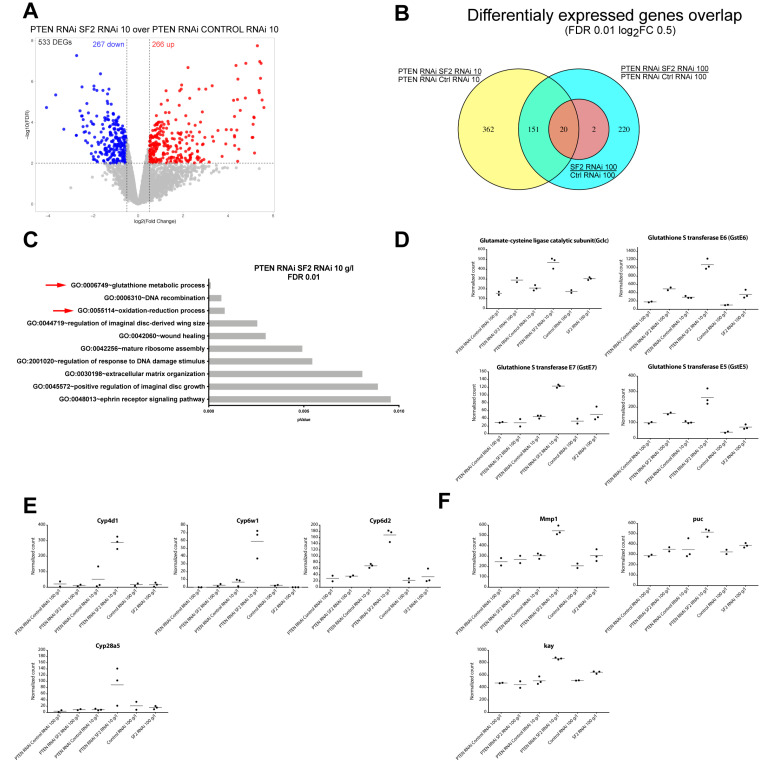
DEGs in *Pten SF2*-deficient eye imaginal discs upon NR are connected to cellular stress. (**A**) DEGs in PTEN RNAi SF2 RNAi discs in 10 g/L conditions. (**B**) Overlap of DEGs detected in PTEN RNAi SF2 RNAi 10 g/L, PTEN RNAi SF2 RNAi 100 g/L and SF2 RNAi 100 g/L. (**C**) GO enrichment analysis of all DEGs (FDR 0.01) in PTEN RNAi SF2 RNAi 10 g/L. (**D**) Chosen DEGs from GO category “glutathione metabolism”. (**E**) Chosen DEGs from GO category “oxidation-reduction process”. (**F**) Downstream targets of JNK.

**Figure 6 ijms-21-04465-f006:**
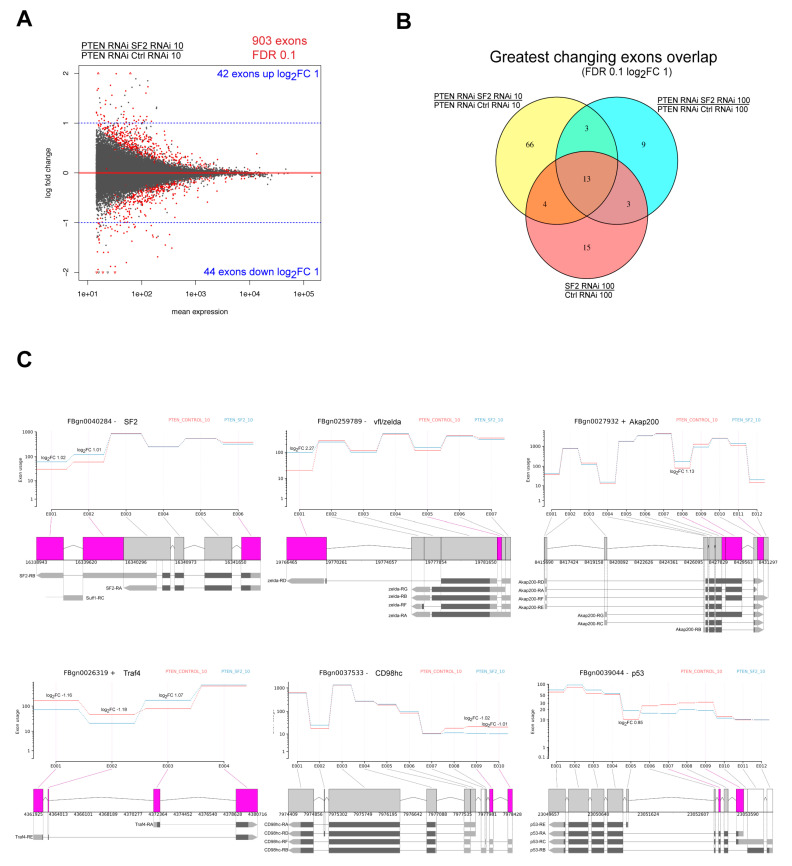
SF2 splicing targets in *Pten*-deficient tissues upon NR. (**A**) Upregulated and downregulated exons in *Pten*-deficient *SF2*-deficient eye imaginal discs upon NR (FDR 0.1). (**B**) Overlap of genes with exon changes detected in PTEN RNAi SF2 RNAi 10 g/L PTEN RNAi SF2 RNAi 100 g/L and SF2 RNAi 100 g/L. (**C**) Chosen examples of SF2 splicing targets among top hits (FDR 0.1, |log_2_FC| 1). Pink shading shows changes with FDR <0.1. Note strand orientation, “+” or “−” after FBgn identifier for gene direction. DEXSeq does not detect exons per se, but “counting bins”, where an exon of variable length gets split into two bins (e.g., vertical line between “E002” and “E003” in SF2 plot), and does not consider overlapping genes separately (e.g., in SF2 plot, what the software depicts as an intron between “E001” and “E002” is actually the first exon of a neighboring gene). Transcript schemes according to FlyBase.

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
