# Peer review of "The Splicing Factor SF2 Is Critical for Hyperproliferation and Survival in a TORC1-Dependent Model of Early Tumorigenesis in Drosophila"

_ijms, 2020, doi:10.3390/ijms21124465_

Round 1

Reviewer 1 Report

This manuscript entitled,  “The splicing factor SF2 is critical for hyperproliferation and survival in a TORC1-dependent model of early tumorigenesis in Drosophila” by Parniewska et al., attempted to identify in vivo regulators of TORC1 mediated overgrowth of Drosophila under nutrient restriction conditions. The authors found a candidate gene SF2, which when knocked down could repress the overgrowth phenotype, and they further investigated the cellular function and downstream targets of this gene. While the findings are of interest, some concerns do exist and need to be addressed.

Major:

  1. The authors chose SF2 based on its involvement in TORC1 signaling (the human SRSF1 possesses S6K phosphorylation sites which are not conserved in SF2). The mRNA and protein levels of SF2 are not changed in PTEN-deficient cells. Thus, there’s not any evidence that SF2 is regulated by TORC1. SF2 regulating the overgrowth phenotype caused by either PTEN/TSC deletion or Rheb over expression does not necessary mean SF2 is regulated by TORC1. SF2 could just independently regulate cell growth caused by different upstream deregulation. The authors stated (line 279) “… likely the activity of SF2 that responds  to the Pten status and therefore TORC1 activity” , and  (in line 290) "its activity is controlled via the TORC1-SRPK axis”. There’s no evidence that can support these statements. 
  2. Along this line, the authors appear to imply that the activation of TORC1 (through deletion of PTEN here) regulates SF2 splicing activity and thus cause the overgrowth phenotype. If that’s the hypothesis, the authors should test if PTEN deletion/mutation caused any aberrant gene expression or splicing events (figure 5 and 6) that is mediated by SF2, which can be further reversed by knockdown of SF2. However, the gene expression changes seem to only happen when knocking down SF2; in other words, genes increased there (eg. Gclc) are not decreased in PTEN deficient cells in the first place. So, these changes can’t explain the overgrowth phenotype caused by PTEN deficiency in the first place. For splicing, the authors didn’t compare these splicing variations between wild type and PTEN deficient cells.
  3. For the splicing variations, most of the examples showed here are UTR regions, which shouldn't alter protein functions. They most likely affect mRNA expression levels. However, the authors claim that their’s not much overlap between mRNA change and splicing change. Then how do these splicing changes regulate the overgrowth phenotype independent of mRNA expression regulation?
  4. Figure1 A-D: siRNA mediated double knockdown was performed to screen candidate genes. The rationale is to find some candidate genes that repress the “PTEN overgrowth”. However, the authors used PTEN knockdown as their control. Then, it’s not clear in this system what’s the phenotype of PTEN knockdown in the first place. A wild-type should be provided here instead. 
  5. Along this line, what is the PTEN overgrowth phenotype? Is it the 1.14 vs 1 upon NR in figure 1C? The authors cited their previous paper (eLife 2013) for the PTEN overgrowth phenotype (line 58), but in that paper (Figure 2D’) they used PTEN mutant, and there’s no difference of eye size in PTEN mutant between normal and NR conditions.
  1. Figure 1C, in the double knockdown system (Pten+siRNA), there are several other candidates (34788 Mo25 etc.) showed significant decrease in eye size under nutrient restrict condition compare to normal condition, maybe mention why would not validate those targets? Are there any positive controls (known to abrogate Pten-driven overgrowth when knocked down) to make sure this read-out is reliable?
  2. 1(C) - Mo25 significantly suppresses Pten overgrowth like SF2. Does Mo25 not affect eye discs of Drosophila
  3. Line 110 the authors stated "Whereas Pten-deficient eye imaginal discs increased in size throughout development under NR… (Figure 2)”, but figure 2 doesn’t show wild-type under NR as a control. “discs deficient for both Pten and SF2 were smaller”, double knockdown is smaller than PTEN single knockdown under NR, but how is it compared to wild-type?
  1. Apoptosis in Figure 2 need to be quantified. Knockdown of SF2 alone in wild-type background under normal condition seems to also have more apoptosis than control RNAi? But in Figure 1E it doesn’t have an effect on eye size?
  2. Line 170: the authors investigated TORC1 activity in TSC mutant background with or without knockdown of SF2. Why don’t they use PTEN knockdown or mutant flies since they should have similarly high TORC1 activity and are more relevant with this study?
  1. It would be helpful if parallel biochemical analysis (Western blot etc.) can be shown alongside the imaging figures. Since author mentioned SF2 could be a direct substrate of TORC1, it would be supportive if the author could show phospho-SF2 level corelates with TORC1 activity.
  2. Authors show that TSC1 mutant cells with SF2 knockdown trigger apoptosis independent S6K activity upon NR. Apoptosis and autophagy are genetically related, and its molecular mechanism is well known. Do eye discs of Drosophila affect autophagy in TSC1/SF2 knockdown tissue? Pten loss causes overgrowth of imaginal discs. Does Pten knockdown cells with TSC1 mutant influence growth of eye discs of Drosophila?

Minor:

  1. In figure 1B, double knockdown of 124 genes with PTEN caused lethality. Are these lines lethal on their own, or are they only lethal when PTEN is knocked down in eyes?
  2. line119: define AEL.
  3. line 81: define VDRC.
  4. Figure 3: what are green and magenta markers? What are the bars on the right and bottom of each figure? Scale bars are too fuzzy.
  5. supplementary figure 8: quantify the difference. wild-type control is needed.
  6. supplementary figure 9a not cited.
  7. supplementary figure 10: define AS. altered splicing??
  8. line 212-214: 533 DEGs... 393 DEGs… These numbers don’t agree with the numbers in supplementary figure 10a.
  9. line248: figure 5c should be 6c.
  1. It would be helpful if author can draw out an overall model.
  2. Try to put scale bar in every image, so the picture is comparable.
  3. “(a) Experimental design” in 1 legend do not mark in bold.
  4. Authors described “21 of 296 knockdowns resulted in suppression of Pten overgrowth” in line 83, but Fig. 1(B) diagram shows 20 genes of suppression. It should be clarified. Authors selected 296 candidates of suppression of Pten overgrowth. It is necessary to state what selection criteria was applied for candidates in this manuscript.
  5. Pten-deficient eye imaginal discs.” in Fig. 2 legend title should be marked in bold. What is AEL stand for? It should be stated in legend or manuscript.
  6. “(a) SF2 knockdown in Pten117 mutant cells (generated by the twin MARCM technique) results in massive apoptosis of the Pten mutant tissue as visualized by cleaved-Dcp1 staining upon NR (lower panel).” in Fig. 3 legend do not mark in bold.

Reviewer 2 Report

A comprehensively thorough experimental study in a Drosophila model that provides compelling evidence for the potential of SFSR1 as a therapeutic target.

I think while the work has strong merit, given the complexity of the pathways involved, and previous issues with developing therapeutics targeted directly at TORC1, then the work would have been strengthened by addition of a translational in experiment in a human cell line demonstrating the SFSR1 knockdown exhibits similar effects in the human cells as the SF2 does in the Drosophila model.

Author Response

We are pleased that reviewer 2 liked our manuscript. He or she is perfectly right that experiments in human cell lines may strengthen our work. However, setting up such experiments is challenging in a lab focused on Drosophila genetics, and they would require a much longer time frame. In our opinion, the demonstration that Pten mutant cells are dependent on SF2 function for their survival in a genetic model already makes a strong case.

Round 2

Reviewer 1 Report

While this manuscript has improved after the revision, there’re still concerns that need to be addressed.   major #1 and 2: These drawbacks of this study should be clearly discussed in the “Discussion” part. Why would control NR samples too small to collect for RNA-seq? According to Figure 1E, the eye size is around 80% under NR compared to control.   major #4 and 5: Please add a figure for the data stated here. For example, a bar plot including: 1) normal condition control; 2) NR control; 3) normal condition PTEN knockdown; 4) NR PTEN knockdown. All of them put together to allow side-by-side comparison. Only normalized to one condition.   major #8: To assess the effect of SF2 knockdown on the overgrowth phenotype of PTEN knockdown cells, it is important to not only make a simple conclusion (higher or lower comparing with or without SF2), but also of biological  significance to have a sense of how much does it slow down the overgrowth, by comparing to a wild type control. Given PTEN knockdown discs are larger, does SF2 knockdown bring the size back to the wild type level? Or is it even smaller than the wild type level? Or it’s actually still much larger than wild type, although it’s smaller than PTEN knockdown? Without a wild type control, the conclusion made only based on PTEN-deficient lines could be misleading.   major #11: First sentence got ignored by the authors.     minor #5 Supplementary Figure 8:  All 6 conditions need to be compared together in one single bar plot. It’s not clear how the statistics were done (number per group, test method, what the ‘*’ symbols represent), which should be included in figure legend. Although it says in “Statistical analysis” that “12 individuals were measured for each genotype”, it appears to be not the case for some of the figures (for example figure 2E).   minor #8 Figure 5A and Supplementary Figure10b: why are genes labeled red (“significants” in figure legend) when they don’t pass the threshold?   minor #10 A model would be helpful to deliver the “simple” key message. For example, what’s happening under 4 different scenarios: Normal condition, wt PTEN; NR condition, wt PTEN; Normal condition, mutant PTEN; NR condition, mutant PTEN. What’s SF2 doing in each condition and how that’s regulated by PTEN and NR?    

Author Response

We thank reviewer 1 for identifying additional weaknesses in our manuscript. Most of reviewer 1’s concerns have been addressed in the re-revised version.

 While this manuscript has improved after the revision, there’re still concerns that need to be addressed.

major #1 and 2: These drawbacks of this study should be clearly discussed in the “Discussion” part. Why would control NR samples too small to collect for RNA-seq? According to Figure 1E, the eye size is around 80% under NR compared to control.

Our manuscript fails to fully depict the challenges of NR experiments. NR not only results in smaller tissues and organs but it also delays development considerably. Unfortunately, the delay is accompanied by a massive spread in initiation of pupariation. Whereas under normal conditions larvae pupariate within a reasonably small time window, NR causes a rather variable delay in pupariation (which is further affected by Pten knockdown, see below). We therefore had to carefully adjust the timing of each experiment, making sure that discs across conditions were at the same developmental stage (to avoid gene expression changes due to developmental timing differences). We focused on eye discs with a large portion of cells still in the proliferative phase (as opposed to the differentiating cells posterior to the morphogenetic furrow). Thus, the eye discs had to be dissected at a relatively early timepoint. Upon NR, the discs without Pten knockdown were so small at this stage that we did not manage to get enough material for an RNA-seq analysis. We now clearly state this in the Results and in the Materials and Methods.

major #4 and 5: Please add a figure for the data stated here. For example, a bar plot including: 1) normal condition control; 2) NR control; 3) normal condition PTEN knockdown; 4) NR PTEN knockdown. All of them put together to allow side-by-side comparison. Only normalized to one condition.

As suggested by reviewer 1, we display the comparisons as bar plots in Figure 1a.

major #8: To assess the effect of SF2 knockdown on the overgrowth phenotype of PTEN knockdown cells, it is important to not only make a simple conclusion (higher or lower comparing with or without SF2), but also of biological  significance to have a sense of how much does it slow down the overgrowth, by comparing to a wild type control. Given PTEN knockdown discs are larger, does SF2 knockdown bring the size back to the wild type level? Or is it even smaller than the wild type level? Or it’s actually still much larger than wild type, although it’s smaller than PTEN knockdown? Without a wild type control, the conclusion made only based on PTEN-deficient lines could be misleading.

We agree with reviewer 1 that a direct comparison with wild-type discs may reveal additional information. However, as pointed out previously, we focused here on the suppressive effect of SF2 knockdown on the Pten overgrowth upon NR. Therefore, we do not have a complete dataset for the wild-type background. The comparison would anyway be tricky because Pten knockdown discs and wild-type discs do not display the same growth behavior upon NR. Quite consistently, wild-type larvae pupariate slightly earlier than larvae bearing Pten knockdowns (even though the knockdown is under ey control; we do not have a good explanation for that). At 156 hrs AEL, there are many prepupae for the wild-type control - but not for the Pten knockdowns – upon NR. The remaining larvae are thus not representative.

major #11: First sentence got ignored by the authors.

It was not our intention to ignore reviewer 1’s comment but we interpreted the statement as referring to Western blots probing for phospho-SF2. In our images, we do show immunofluorescence of cleaved Dcp-1 and phospho-S6. In both cases, the localization of the signals is crucial. We therefore do not think that Western blots (averaging dozens of imaginal discs) would add important information.

minor #5 Supplementary Figure 8:  All 6 conditions need to be compared together in one single bar plot. It’s not clear how the statistics were done (number per group, test method, what the ‘*’ symbols represent), which should be included in figure legend. Although it says in “Statistical analysis” that “12 individuals were measured for each genotype”, it appears to be not the case for some of the figures (for example figure 2E).

We now compare all conditions in bar plots, as suggested. We thank reviewer 1 for pointing to the fact that in some statistical analyses, n deviates from 12. We now state this in the Materials and Methods. (This is actually the case for statistical analyses in imaginal discs. For adult eyes, n is always 12.) All tests and symbols (*) are used consistently throughout the manuscript. Therefore, we describe the statistical analyses in the Materials and Methods rather than in each figure legend.

minor #8 Figure 5A and Supplementary Figure10b: why are genes labeled red (“significants” in figure legend) when they don’t pass the threshold?

We have changed the volcano plots in Figure 5a and Supplementary Figure 10b such that the significantly downregulated genes are displayed in blue and the significantly upregulated genes in red. The corresponding numbers are also indicated in the respective colors.

minor #10 A model would be helpful to deliver the “simple” key message. For example, what’s happening under 4 different scenarios: Normal condition, wt PTEN; NR condition, wt PTEN; Normal condition, mutant PTEN; NR condition, mutant PTEN. What’s SF2 doing in each condition and how that’s regulated by PTEN and NR?

A schematic representation of all eight scenarios for both organ size (eye disc) and clone size is shown in Supplementary Figure 11. As we state in the manuscript, it is presently unclear whether and how SF2 is regulated by Pten and NR.

Reviewer 2 Report

Whilst I appreciate that it would take more time to include the human cell line data, and that the laboratory of the authors is focused on Drosophila genetics, there are plenty of collaborative partners (and even commercial services) available that could perform the knockdown in a human cancer cell line in a relatively short time frame.

Therefore I still feel that this is a sticking point to publication in IJMS

Author Response

The suggested experiments are certainly of interest but cannot be performed within the time window for the revision.

In addition, there is a caveat to the cell culture experiments. The most prominent phenotype we observed upon SF2knockdown is the elimination of Pten clones under NR conditions. Importantly, this effect fully depends on a clonal situation. It will be very difficult to achieve a similar context in a cell culture system. One would probably have to use MDCK cells capable of forming epithelial structures. However, we would have to establish a co-culture system for two genetically manipulated and differentially marked cell populations. This is an interesting option but not straightforward and certainly beyond the scope of the present study. Furthermore, even with this elaborated system, we would still not be using human cells.

Exploiting Drosophila genetics allowed us to assess SF2’s function in a clonal context in vivo. We would like to argue that this is exactly the strength of our study.